# Satirizing News Media, Changing Taiwan's Feelings: *The Night Night Show with Brian Tseng*'s Adaptation of the American Satire News Format

Muyun Zhou

Departments of Asian Studies and Comparative Literature, Pennsylvania State University, State College, PA 16801, USA; mqz5352@psu.edu

**Abstract:** The US television program *The Daily Show* has inspired creative talents worldwide to adapt the American political satire news formats to their own political environments. One example is *The Night Night Show*, hosted by Brian Tseng between 2018 and 2020 and produced by the STR Network from Taiwan. Instead of approaching the show as the result of the diffusion of the US cultural and political model into the rest of the world, this article contextualizes *The Night Night Show*'s adaptation of an American satirical news format in the Sinophone political discourse of laughter and satire in the modern history of Taiwan. It argues that while the show's adaptation of an American satirical news format demonstrates how satire can dismantle linguistic and national boundaries as a transnational bonding force, it also brings this American format to critical scrutiny. In particular, the principal cultural understanding of news media as sensationalistic and propagandist instead of truthful in the local context contests the notions of "truthiness" central to the American satire news formats.

**Keywords:** Sinophone political satire; truthiness; news media history; sensationalist aesthetics

## 1. Introduction

Ask any youth from Taiwan on the street. The person might not have sat in front of the TV to watch a news report for a while, but they are a lot more likely to have seen those news reports being trolled ruthlessly as they were scrolling YouTube on their phones and bumping into a trending clip from *The Night Night Show*. *The Night Night Show with Brian Tseng* (*boen yeyexiu* 博恩夜夜秀, a.k.a. *The Night Night Show*) is a late-night satire news talk show on the STR Network in Taiwan. Hosted by Brian Tseng (曾博恩) and his standup comedian colleagues, *The Night Night Show* is Taiwan's first political satire talk show presented in the American satire news format. The show not only has a ticketed live audience but also circulates its episodes online for free, through compiled segments on the STR Network's YouTube channel. *The Night Night Show* began when the host Brian Tseng and his former employer DJ Hauer (謝政豪) started the STR Network—STR as for "satire"—in July 2018, when they noticed an absence of political satire shows like *The Daily Show* and *Last Week Tonight* from American TV in Taiwan despite the overwhelming interests in political and social topics in public (Yeh 2018). They then debuted *The Night Night Show* in August 2018. The show was an immediate success, with its pilot episode drawing over 130,000 views in five days (Liu 2019).

In an article titled "Life of Brian: the comedian telling China where to go", *The Economist* hails *The Night Night Show* as a milestone in the history of Taiwan comedy (Nguyen-Okwu 2019). Citing Dr. Chen Sy-shyan, a professor of political philosophy at National Taiwan University, the article claims that Taiwan has been an "island of tragedy" that "always lacked a tradition of comedy, with no environment or culture to foster a sense of humour" since 1949. In contrast, Brian Tseng's political satire filled the vacancy of comedy and joy in Taiwan by borrowing and adapting the US satire news format to Taiwan. While it is important to credit *The Night Night Show* for pioneering its particular format, the type of comment made in this

*Economist* article reinforces a diffusion model of democracy (O'Loughlin et al. 1998) as it is defined and created by the US and spread into other spaces in the world. The agency of political and cultural actors from Taiwan is pushed to the background.

To re-center the local actors like Brian Tseng, there needs to be a discussion on how satire news in Taiwan problematizes the concerns of the US news satire format to resituate *The Night Night Show* within the history of Sinophone comedy, satire, and politics. Instead of using the show as an example of how American satire news is saving Taiwan, or even "telling China where to go", why does Taiwan need American satire news in the first place? In this article, I will engage with *The Night Night Show* at three levels. First, I take the different histories of news media's development and their different social dynamics seriously to reconsider if, how, and where the central problem of "truthiness" in the American satire news genre can be found in Taiwan. Second, I ask what events and lineages of Taiwan's multimedia satire culture drew its youths to a foreign format in 2018. In particular, I will situate the show in the aftermath of the 2014 Sunflower Movement. Third, I analyze *The Night Night Show* through its key segment "Ay!" to observe how Brian Tseng absorbed but exceeded the American satire news format. I conclude with a discussion about the blowback against the show and the critical legacy the show has left for the Sinophone satire news' future.

## 2. Locating the Problem of "Truthiness" in Taiwan News Media

American satirists have been foremostly concerned with the discourse of truth that the news media and politicians have advanced, and the various formats of "satire news" are the principal location where such concern is expressed. Satire news generally refers to a TV entertainment genre that parodies mainstream journalism, either by making satirical comments or comedy sketches on actual news events or by presenting fictional news stories. In 2005, the American TV satirist Stephen Colbert coined the term "truthiness" in the premiere episode of his late-night talk show, *The Colbert Report*, which features a satire news segment itself. "Truthiness" according to Colbert emerged at the height of the anti-rational media environment from both the left-wing and the right-wing discussions under George W. Bush's presidency. As a neologism, the term signifies situations when "truth" means something one feels or wishes to be true, rather than something that is grounded in facts, evidence, or reasons. On the cultural-political level, "truthiness" is also a critical lens through which one can observe how the discourse of truth is facilitated by media play and manifests in public rhetoric.

Through "truthiness", scholars have suggested considering the satire news genre as an innovative political intervention in the US context. Against the instinct that satire news risks turning real-life politics into comic entertainment and fostering political apathy or cynicism, studies argue instead that satire news can incentivize the viewers to challenge the presentation of social issues in mainstream news media (Baym 2009; Jones 2009; Day 2011; McClennen and Maisel 2014; Caron 2021). Among these studies, many have rightfully echoed Geoffrey Baym's articulation of how the public rhetoric of truth has created both "the conditions *and the need*" for satire programs on TV to emerge as political site (Baym 2008). Not only can satire news programs effectively dissect political dilemmas and encourage their solutions in addition to generating humor, but the American audience is also enabled by these programs to supplement the political discourse with comic aesthetics, to participate in civil affairs via popular media texts, or to engage in progressive political actions. And yet, with their commitment to an archetypal news media as truth-bearing sites, underlying these studies on US-based satire news is also an opposition between truth and truthiness, or between reasons and passions: the former implies constructive criticism and democratic responsibility, while the latter suggests sensationalized fabrications and authorial interests.

In the case of Taiwan, the consideration of news media as truth-bearing sites is not as vital as the desire for them to address concerns with authoritarian propaganda or sensationalist tabloidization. This different order of priority is inseparable from Taiwan's media

history in the martial law (1949–1987) and post-martial law periods (1987-present).[1] During the martial law period, although the Republic of China (ROC) government in Taiwan aligned with the American Free World order during the global Cold War, news media in Taiwan was neither democratic nor free. There were only three legally run TV stations in Taiwan, known as the "Old Three" (*laosantai* 老三台). All were terrestrial broadcast TV stations, and each was run by one of the three branches of the state: the Taiwan Provincial Government, the Chinese Nationalist Party (KMT), and the Ministry of National Defense. In that period, the media industry was understood as an instrument for political propaganda under the authoritarian KMT regime. Its goals were to aid the KMT in negotiating its path to a KMT-centered Chinese nationalism, to the sovereign entity of China against the People's Republic of China (PRC) as its rival, and to a cultural-cum-ideological war waged against global Communism with its Free World allies.

When martial law was lifted in 1987, Taiwan's news media underwent rapid liberalization and democratization. The end of martial law was followed by the press ban being lifted and the "Removing Political Parties, the Government, and the Military from the Old Three Movement" (*dangzhengjun tuichu santai yundong* 黨政軍退出三台運動) that called for marketization, deregulation, and democratization of Taiwan's media industry. In this post-martial law period, newspapers, magazines, and satellite and cable TV exponentially grew. In the TV industry, the illegal and non-partisan cable TV stations that emerged at the end of the martial law period—also known as the "Fourth Station" (*disitai* 第四台)—became legalized in 1993. The "Old Three" also went through privatization and became corporatized (Department of Policy Research and Development 綜合規劃處 2016). However, this urgent transition from party-driven authoritarianism to a market-driven neoliberal democracy only resulted in a "media disorder" (*meiti luanxiang* 媒體亂象) in the late 1990s. The term was created to describe the Taiwan media industry which became filled with sensationalist and low-quality news and talk shows (Wei 2018). Much less democratic free speech or truth, propagandist interests took a backseat to the Taiwan TV industry as market-driven interests dominated.

Through their scholarship on the "media disorder", Taiwan's communication scholars have provided various explanations for its media's failure to become truly democratized post-martial law, where conversations on facts, evidence, and reason would be possible. Examining the first twenty years of Taiwan's post-martial law transition across print and broadcast media, Shih-Hung Lo expresses that the Taiwan media never fully managed to transition out of its clientelist characteristic from the martial law period (Lo 2008). The mainstream newspaper and cable TV stations have continued to thrive through their leadership's political participation with the KMT party and their mutually beneficial relationships with major party members. More concerning than the clientelism for Lo is the damaging effects of the unsupervised commodification of media in Taiwan. He provides examples of malignant competition from imported tabloidization or "Apple-ization" (*pingguo hua* 蘋果化), where news media are forced by market competition to prioritize their entertaining and sensational characters and lower their price. News media also became reliant on campaign advertisement income, which makes their political commentaries superficial and instrumentalized. In Lo's opinion, whereas media used to be the propagandist embodiment of political authorities during the martial law period, they have now become shallow embodiments of neoliberal political entrepreneurs. In contrast, Ti Wei holds a slightly different opinion from Lo in that the disorder "is not caused by neoliberalism but by the immaturity of the democratic system" (Wei 2022). Wei argues that the key reason behind the capitalists' take-over of the previous state-controlled media is the media sectors' lack of competitiveness and market adaptability within themselves and that this problem cannot be easily solved by shifting the media's ownership from private to public. He also remarks the survival threat to print and broadcast media today should not be completely traced to the malignant market competition, as their primary challenge is the rise of transnational social media and digital platforms.

Although the accounts for the "media disorder" differ, the opposition between truth and truthiness, or between reason and passion persists. However, in the case of Taiwan, the emphasis is shifted to the indispensability of the latter. In both Lo and Wei's analyses, for example, sensationalism and tabloidization characterize media's "disorder" and diagnose its embeddedness in the neoliberal economy, either because of the media sectors' immaturity in defending themselves against it, or because of the strength of such economic-cum-political force to take over Taiwan's media sectors. Conversely, especially prominent in Ti's analysis but gestured by both is how this "media disorder" is not only obtrusive to news media's access to the democratized public rhetoric of truth but also as *normal and necessary* in pursuit of an anti-authoritarian national body post-martial law. Then, if the point of parodying the mainstream news' truthiness in American satire news TV is about how intellectual and rational political conversations over truth can be both fun and critical, satirists from Taiwan like Brian Tseng are tasked with a more radical change: If this fun, as it is epitomized in the current omnipresence of tabloid aesthetics in Taiwan new media, is not a "can" but a "must", what are the options left for the satire news media and their audience to imagine their rational access to public rhetoric of truth?

## 3. Why Does Taiwan Need American Satire News?

In Taiwan, *The Night Night Show* belongs to a wave of political satire shows that emerged in the aftermath of the Sunflower Movement of 2014. On 18 March 2014, a coalition of students and civic groups in Taiwan temporarily seized the Legislative Yuan during a joint session while protesting a free trade agreement with the PRC, the Cross-Strait Service Trade Agreement (CSSTA). Then-President Ma Ying-jeou from the KMT party prepared to force through the agreement without having it reviewed by the opposition DPP, as previously agreed. Sociologist Anson Au (2017) points out the CSSTA's significance is political, not economic. It is widely understood today that the Sunflower Movement held a specific complaint about the KMT's black box politics and led to the party's failure in the 2016 election. Au further observes that the movement is consistent with the burgeoning anti-Sinoism in general within the society of Taiwan. Anti-Sinoism was originally a bottom-up sentiment from the martial law period, posited against the KMT's policies that reinstated an orthodox Chinese identity across Taiwan at the expense of Taiwan's indigenous population and pre-1949 residents' interests.[2] In Au's opinion, the Sunflower Movement rearticulated this martial law period sentiment into a civic nationalism in 2014, which defines a national identity by civic participation and contribution rather than by ethnicity, place of origin, or cultural heritage.

The Sunflower Movement's impact on youth culture is twofold. On one hand, as the sociologist has suggested, the Sunflower Movement showcases the multifarious channels of confluence between party politics and grassroots activism in Taiwan (Ho 2018).[3] The movement extended both the KMT and DPP's youth and activist outreach, with some key participants absorbed into the DPP post-movement. On the other hand, the movement drove enthusiasm among young people in Taiwan for independent media to express their civic responsibility on the media front. Youths have chosen to launch their own media instead of becoming journalists under mainstream media platforms. One popular direction is for civil society groups or interested activists to launch independent fact-checking initiatives against misinformation and disinformation in Taiwan news media (Rawnsley et al. 2022). Another rising trend is sponsor-free or crowdfunded journalism, although no media can be perfectly independent of business or political connections. For instance, members of the Sunflower Movement were hired by a newly founded ad-free news site, *The Reporter*, in 2015 (Hérait 2022). The non-profit media outlet can bypass the limit of journalistic depth placed by political campaigns and business sponsorship by gathering micro-donations from public fundraising efforts. More journalism fundraising platforms were also launched in response to the rise of crowdfunded journalism in addition to existing ones (Rawnsley et al. 2022).

This turn to independent media and the growing crowdfunding infrastructure together created the appropriate environment for political satirists like Brian Tseng to arise.

Before founding the STR network and starting *The Night Night Show*, Tseng worked as a producer and a writer for the education YouTube channel Taiwan Bar. It was launched in the same year as the Sunflower Movement to produce an animation series on the history of Taiwan that begins with the Japanese colonial period of Taiwan instead of KMT's 1949 retreat. Undoubtedly funny but not always satirical, Taiwan Bar was among the early iterations of a "national Taiwan" identity that challenged the KMT-centric history on independent media platforms. Other independent political satire programs have also been on the rise since 2014 in Taiwan. For instance, a YouTube satire news channel called EyeCTV, or "Central Eyeball Television" (*yanqiu zhongyang dianshitai* 眼球中央電視台), was started in 2016. In its flagship show *EyeCTV Xinwen Lianbo* (眼球中央電視台新聞聯播; lit. trans. EyeCTV News Simulcast), the EyeCTV comments on contemporary news, produces parodies of the PRC's national *CCTV Xinwen Lianbo*, and calls itself "the national TV station of the ROC". As such, it equates the implicit propagandist rhetoric in Taiwan's neoliberal media discourse to the explicit propaganda in mainland Chinese media and satirizes the gap between the ROC's constitutional vision of Taiwan and its practical reality (Lee 2019). With 1.18 million subscribers and between 300,000 and millions of views per episode on YouTube, EyeCTV runs on income from ad collaborations, merchandising, and cultural funding in addition to its YouTube revenue (Rother 2022). In 2018, Brian Tseng left Taiwan Bar and started the STR Network and *The Night Night Show* with the co-founder of Taiwan Bar, DJ Hauer. The production moved from exclusive crowdfunding to an array of funding sources later, such as audience ticketing, advertisement, merchandising, guest promotional fees, and cable TV and OOT-oriented broadcasting rights (Chen 2018). Here, both the *EyeCTV Xinwen Lianbo* and *The Night Night Show* are only the tip of the iceberg of Taiwan's satire news programs that emerged through American commercial platforms YouTube and Apple Podcast after the Sunflower Movement. Across the board, however, these programs fit into Taiwan's mediascape that expresses an anti-Sinoist sentiment and a wish to break away from the political and financial restrictions of mainstream platforms by pursuing crowdfunded programs.

Within this new wave of Taiwan's political satire programs, *The Night Night Show* was neither the first nor the only program that introduced the satire news genre to Taiwan's contemporary political landscape. However, Brian Tseng's choice of the American satire news format over the many available local variants should be carefully examined as an intentional choice. One of the reasons could be Tseng's previous experience with standup comedy (*tuokouxiu* 脫口秀), a format first known to Taiwan through US TV talk shows (Social 張碩修 2021). In the Sinophone world, the standup format first emerged among the expatriate community in Hong Kong and was performed in English. The Cantonese standup scene began in the 1990s (Tam 2018). Then, in the 2000s, the standup format was popularized across the Sinophone communities in the US, Taiwan, and mainland China (Tam 2018; Social 張碩修 2021, p. 324). In Taiwan, the first standup comedy club "Live Comedy Club Taipei" (*kamidi xiju julebu* 卡米地喜劇俱樂部) was founded in May 2007 by the former theater director Social (張碩修) in a basement on the Taishun Street in Taipei (Social 張碩修 2021, p. 256). Brian Tseng first learned about the standup format during open-mic nights in college, but he debuted as a standup comedian in 2016 at the Live Comedy Club Taipei by winning its competition (Social 張碩修 2021, p. 296). In 2017, Tseng uploaded a video of his performance at the club, titled "*da nai wei wei*" (大奶微微; lit. trans. big-milk-little-little), onto his YouTube channel. This video made Tseng a pioneer in that he was one of the first comedians in Taiwan who was willing to share their works on public media platforms. Whereas standup comedy has been widely performed on national TV in mainland China since 2012, most comedians in Taiwan hesitated to put their works on the media platforms as they worried those platforms' free streaming services would devalue their live performances (Deng 2022). That year, Tseng's "*da nai wei wei*" video went viral and gained him a significant YouTube following. Building on this YouTube success, Tseng's satire news career at the STR network began.

*The Night Night Show*'s unique American format also juxtaposed itself locally against a long 20th-century history of satire culture in Taiwan, which the show has carefully denounced. Contrary to *The Economist*'s portrayal of Taiwan as "an island of tragedy", previous studies have shown that neither during nor after the martial law period is Taiwan ever devoid of engaged and critical comedic forms. In her comparative study of Taiwan and South Korea during the Cold War, for instance, the literary scholar Evelyn Shih (2018) observes a comic aesthetic of caricature, genre games, and nonsense that flourished in Taiwan under martial law across print media, radio, film, and television. Escaping censorship, these comic aesthetic forms "playfully elaborate negativity" toward the KMT state's anti-Communist patriotism and perfunctory embrace of the American Free World order (8). In the post-martial law period, the politically engaged comedy became the most prominent in performance arts, especially via experiments with the traditional Chinese verbal art of crosstalk (*xiangsheng* 相聲) that carries a satirical (*fengci* 諷刺) function (Luo 2008; Moser 2018).[4] Notably, in 1984, a group of second-generation mainland émigrés (*waishengren* 外省人) from the theater circles, including the Washington D.C.-born returnee Stan Lai, formed the Performance Workshop (*biaoyan gongzuofang* 表演工作坊) and invented the genre of "crosstalk plays" (*xiangsheng ju* 相聲劇), "performed in the mimicry style of *xiangsheng*" (Guan Lim 2015, p. 73). The crosstalk plays retained crosstalk's function of political satire. For example, in *Taiwan* Gaitan (Strange Talks About Taiwan/台灣怪談; 1991), which was revised from the solo crosstalk format, the comedian Li Liqun acting in-character brought political news under attack, such as the broadcasted fights during Legislative Yuan (Taiwan's legislature) meetings, the Protecting the Diaoyu Island Movement, and the smoke-free bans.

With the 2014 turn toward anti-Sinoist civic nationalism in Taiwan, the American satire news format signals a mindful break from the previous political satire traditions, especially the crosstalk revival. Since its early days, the Performance Workshop conceived its crosstalk plays in dialog with American standup comedy. In an interview with *Los Angeles Times* (Epstein 1994), for instance, Stan Lai carefully distinguished the "Western stand-up comedy" from the Performance Workshop's "Chinese stand-up humor": "I have a very shallow understanding of Western stand-up comedy, but that rhythm is basically one setup and one punch line, so Chinese stand-up humor [in crosstalk] unfolds much more slowly and graciously. People (from other cultures) can get used to that rhythm, but it may take some adjusting". Growing up in Cambridge, Massachusetts with his scientist father for seven years and getting two master's degrees from London and Paris, Brian Tseng speaks English like a native and shares Stan Lai's third-culture background, since Lai also spent twelve years in Washington D.C. as a diplomat's child and later obtained his doctorate from University of California Berkeley. However, whereas Lai turns toward his ancestral homeland by adapting the crosstalk format upon his return, Tseng looks in the opposite direction toward the US. Lai's adaptation of crosstalk has been discussed in the aftermath of the "Native Land Movement" (*xiangtu wenxue yundong* 鄉土文學運動) (Mao 2005; Lim 2015; Wang 2004), where his search for a lost art form from mainland China in Taiwan articulated his generation's uneasy identification with the sense of Chineseness. Brian Tseng's turn toward the US, in contrast, marks a symptom of his generational aphasia, where that imagined bond with an ancient Chinese homeland as a language to articulate the self is decidedly severed for a nascent "Taiwanese"-ness. The faith in the US here comes not so much in its ideological perfection or superiority but in its promise as a new language to articulate an emergent identity.[5]

Moreover, as a televised media extension to live standup performances, the American satire news format of *The Night Night Show* evokes the satire theater of earlier decades in Taiwan, such as the crosstalk plays, where politics and comedy are inseparable. Furthermore, the show takes this union of politics and comedy to intervene in the TV culture in Taiwan, where politics is kept intentionally at a distance from comedy. The period between the end of martial law and the popularization of political satire on independent media in the 2010s is generally considered the golden era of TV comedies in Taiwan. For instance,

one of the most successful Sinophone variety-comedy talk shows, *Kangsi Coming* (*Kangxi laile* 康熙來了), ran on the CTi Variety channel from 2004 to 2016 for a remarkable 12 years. However, in that period, TV variety shows never joked about partisan or Cross-Strait politics, as they aimed to entertain the audience regardless of political affiliations. The early TV comedians commonly used Weibo (Chinese Twitter-now-X) to interact with their mainland viewers to maintain a presence in mainland China's entertainment market. This is not to say Taiwan's TV did not have political shows, but those were kept separate in the many political (call-in) talk shows on TV, which had clear political orientations and loyal viewers, financially supported by commercials and the political party they were aligned with (Lee Chen 2017). Conversely, the generation of comedians like Brian Tseng positioned their media platform on YouTube, which is kept outside the Great Firewall of mainland China. In his abandonment of the massive market across the strait, Brian Tseng preemptively chose his battle: his fight is less about getting into the controversies around the political status of Taiwan in the Cross-Strait relation, as it is about piecing together the severed limbs of politics and comedy on the Taiwan post-martial law media platforms.

### 4. More than a Copy: *The Night Night Show*'s Adaptation of American Satire News

Despite being a recognized pioneer in Taiwan, Brian Tseng has expressed his frustration with viewers calling *The Night Night Show* a copycat of the American TV talk shows (*Tai Sounds* 2018):

> A term that people tease me with is "The Awkward Copycat Show." "The Awkward Copycat Show" very tersely summarizes the negative opinions about *The Night Night Show*. Many people say we are copying Conan, John Oliver, Jimmy Fallon, and even Oprah. […] True, we have watched many videos. Have we been influenced? We must have been. But this doesn't mean that there's nothing original in [the show]. If you start a new format, how can you guarantee its success? A more secure way to go about is to study how successful people achieved their success, and after imitating them for a while, you will find elements you want to add every here and there. There are things only I can do. Gradually, you will find your personal style.[6]

Tseng's response to these accusations of plagiarism vaguely traces out the evolution of his show. When the pilot episode of *The Night Night Show* was uploaded to YouTube, it only had four main segments, most of which could be traced to similar segments in *The Daily Show* with Jon Stewart. *The Night Night Show* opens with a three- to four-minute segment "Random News Report" (*xinwen luanbao* 新聞亂報), where Tseng makes a parody of traditional news broadcast with a set of recent domestic and Taiwan-related news. Then, the show moves into another segment of comparable length, "Correspondent Jim" (*tepaiyuan* Jim 特派員Jim), where Tseng's colleague Jim conducts parodies of journalistic street interviews across Taipei. The segment "Ay!" (*ei!* 欸!) comes after. In the 10 to 20-min-long section, Brian Tseng delves into one recent social news issue and dissects it satirically for the audience. Finally, the show concludes with one or two interviews with Taiwan's politicians and celebrities. Woven these segments together were some on-site or pre-recorded short comedy skits, such as "Better than Brian" with Hello (賀瓏). However, the "Correspondent Jim" segment, originally inspired by *The Daily Show*'s most popular "Correspondents" segments, only lasted for seven episodes. As it went into its final season with Brian Tseng in 2019, the show routinely included skits performed by other standup comedians on the team.[7] Additionally, the segment "Ay!" grew to become the staple segment of the show.

While *The Night Night Show*'s episode format grew more distinct from that of *The Daily Show* across the years, part of the success of the segment "Ay!" can be attributed to its biting humor and analytical depth that mirrors but exceeds the comparable units in US satire news shows. The segment "Ay!" resembles the generic formula of American satire news from late-night talk shows, which combines the host's serious comments with light-hearted jokes, caricatures of political figures, and digressive responses. For instance, a standard

satire news segment of *The Daily Show* with Jon Stewart runs about five to ten minutes and usually features one social issue per segment. Jon Stewart sits in a mock news broadcasting room and refers to a rectangular image area as he introduces the subject. In an actual news program, the images shown in the rectangular area are properly journalistic pictures, stills from a longer video featuring politicians' serious faces, or graphs of visualized data. In contrast, *The Daily Show* with Jon Stewart introduces parody by incorporating about three to five boldly edited graphics per episode alongside a similar amount of video stills but captured at times when politicians have distorted facial expressions. Among the graphics, *The Daily Show* mostly references non-age-specific pop culture or household items in American families. It has also revised the text portions on TV and news media's logos into witty puns when it explicitly criticizes the media's role in presenting a social issue. Memes are not commonly used. In addition, the show inserts official headshots of the politicians when they are under attack. As the segment's narrative goes, the show prioritizes offering important cultural, historical, political, economic, or religious contexts that supplement the issue beyond the aspects presented by mainstream media. Then, with newsreel clips, Stewart points out the inherent fallacies in the news figure's speeches or the inconsistencies between their words and deeds. Nonetheless, the criticism rarely extends into the more complex aspects of systemic failure or global geopolitical impasse, unless it is already in the central conflict of the social issue. Other shows like *The Colbert Report* and *Last Week Tonight* also include skits or guests in their satire news segments. Yet overall, they are comparable in length and in how they incorporate visual elements and offer political criticism.

As he sits behind a wooden office desk in front of an LED screen showing Taipei's skyline, Tseng's allusion to American news satirists is hard to miss. Dressed in a full suit, Tseng engages with the pop-up square window on his left-hand side throughout the segment, showing memes, screenshots, headlines, etc., and cutting scenes between *The Night Night Show*'s production studio and clips from newsreels. In this way, *The Night Night Show* and its American predecessors use the montage technique, as well as the juxtaposition between the comic image and the host's parodic script, to create the comic effect. However, despite their similarities, "Ay!" exceeds the presentations of critical social issues in satire news segments of American late-night shows not only because it is usually twice or thrice as long but also in its two key characteristics: first, it offers well-structured narratives that approach Taiwan's malaise from multiple institutional angles beyond the specificity of the featured social issue, and second, it uses a density of jokes, caricatures, and memes that specifically respond to and participate in the youth and internet culture in Taiwan.

As a case study, I will primarily engage with the last "Ay!" episode uploaded to the STR network's YouTube channel in January 2021 after *The Night Night Show* ended, titled "Ay! Ractopamine Pigs" (*ei! laizhu* 欸! 萊豬), as I see it represents the comic form and political alignment of the show at its most mature stage (Tseng 2021). The over-20-min-long episode focuses on Taiwan's decision to lift the ban on importing pork containing ractopamine, made by the DPP's Tsai Ing-wen government on 28 August 2020. It was widely understood that the decision was made after enduring pressure from the US regarding its meat exportation to the Taiwan market, ever since related restrictions were put in place in 2006 (*BBC News Zhongwen* 2020). Nevertheless, Tsai's lifting of the ban not only triggered concerns about food safety but also got on people's nerves about state jurisdiction, as the Ministry of Health and Welfare did not allow provincial governments to administer a local ban despite the ban's lifting never having passed through the state's Legislative Yuan.

In "Ay! Ractopamine Pigs", Brian Tseng ruthlessly criticizes political figures from the DDP government and the KMT party, while recognizing the systemic political-cum-economic impasse of Taiwan. The episode opens with Tseng's scathing critique of DDP's performative politics. He plays a news clip of the DDP and the KMT members throwing pig guts at each other during a Legislative Yuan meeting, followed by another clip of the then-President of the Executive Yuan, Su Tseng-chang of the DDP, reading off a script "Taiwan is such a blessed land in the turbulent world" despite his face being covered by a protest plaque throughout his speech (Figure 1). Next, Tseng quotes the catchphrase "Campaign

Champion, Governing Goon" (*zhihui xuanju, buhui zhiguo* 只會選舉, 不會治國) to attack DPP's impotence in seeking new policy solutions that can better protect people's welfare than the previous KMT government. He follows the quote with visually exaggerated news clips, such as placing President Tsai's condemnation of the KMT's decision to soften the ractopamine beef ban from 2006 in a split-screen against her recent TV appearance, asking people's understanding of her decision to lift the ban on ractopamine pork (Figure 2).

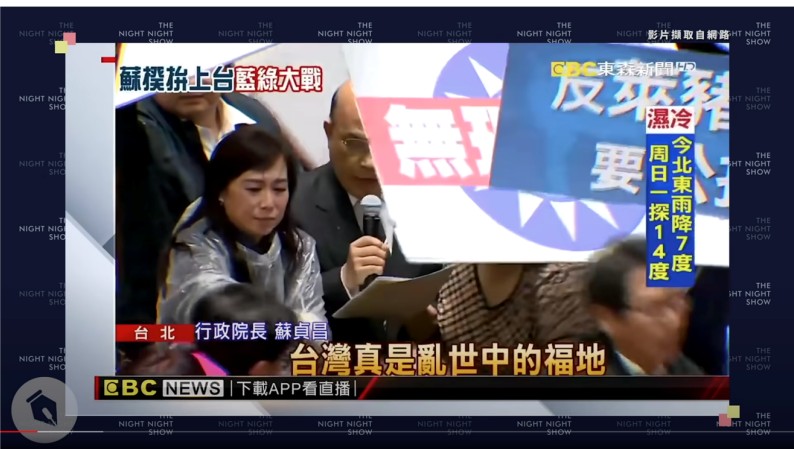

**Figure 1.** Face covered by a protest plaque, Su Tseng-chang is shown announcing "Taiwan is such a blessed land in the turbulent world" (3:34).

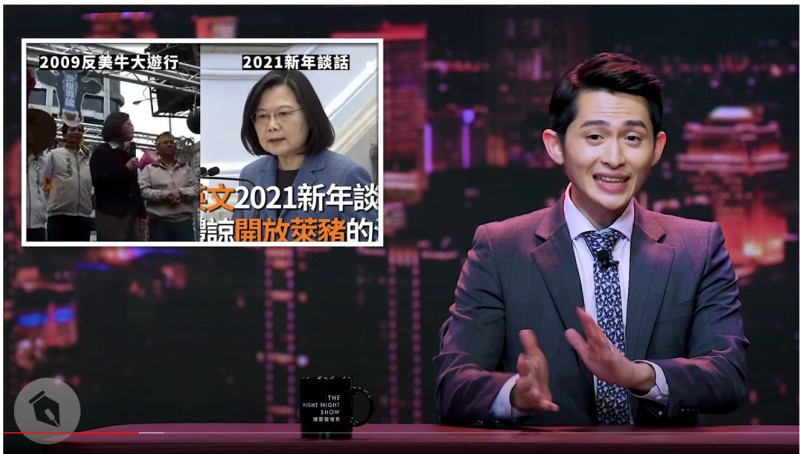

**Figure 2.** Tsai Ing-wen's before- and after-election videos are put into split-screen to heighten the contrast (5:24).

Then, Tseng turns toward the KMT with equal harshness. Tseng plays several news clips featuring former President Ma Ying-Jeou, where Ma goes back and forth between his recent criticism of Tsai's decision as administrative negligence of people's welfare and his previous indulgence in American pork (which is implied to contain ractopamine), as shown through an early video of Ma sharing a "small secret" to his audience about how much he loved eating pork knuckles while he was a student in the US. As such, Brian Tseng astutely suggests that, while it opposes the government's decision, the KMT is equally apathetic toward people's health and welfare and acts out of sheer political gesture as the oppositional party.

In addition to the KMT and the DDP, the Chinese Communist Party (CCP)'s government in mainland China also off-handedly comes under Brian Tseng's attack twice. First, he uses a meme featuring CCP's President Xi Jinping drinking baby formula labeled with melamine. With the melamine recalling a major food safety incident in mainland China

from 2008, Tseng quotes the PRC government's food safety policies to foil Taiwan's more derelict regulations on the matter. Then, to ridicule a KMT legislative member's concern over what the domestic market for the ractopamine pork would be, Tseng suggests that it can be made into pork floss and be exported to mainland China (Figure 3). In contrast, another major player in the ractopamine pork incident—the US government that pressured Taiwan into this decision—is surprisingly left untouched by Brian Tseng's satire.

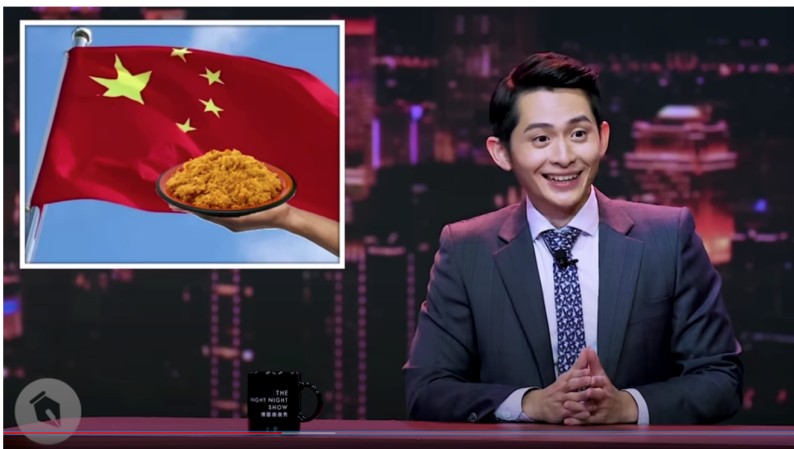

**Figure 3.** Tseng suggests exporting ractopamine-containing pork floss to mainland China (7:53).

In the end, Brian Tseng poses two questions to suggest the problems at the core: first, Taiwan's food safety issues, and second, the international status of Taiwan. It seems that to form international and political alliances, Taiwan will necessarily compromise domestic food safety. As a finishing touch, Tseng brings three Taiwan piglets on stage for a race, where Tseng leaves the piglets to decide whether Taiwan should prioritize its international status or its people's food safety.

In addition to the analytic depth, "Ay!" also exceeds the American satire news as it engages with an unprecedented number of memes, jokes, and caricatures that explicitly align the show with a grassroots political positionality, associated with Taiwan's youth culture rather than family-oriented pop culture in general. Over 20 min, Tseng makes more than 14 verbal and graphic caricatures of politicians across the CCP, DPP, Taiwan Jianguo Union, and the CCP based on their appearances, words, and policies. One repeating motif is the *Tenet* memes, where Tsai Ing-wen and Ma Ing-jeou are made into personalities from the Hollywood film in a repetitive and symmetrical manner, each appearing two times as Tseng criticizes the vacillating policies from both sides (Figure 4). In addition to *Tenet*, the show also makes memes and jokes with other popular references such as the "Coin Master" ads featuring Jennifer Lopez streamed across Taiwan's internet, the anime *Chūka Ichiban!* that almost every youth in Taiwan has watched in childhood, the Marvel character Loki, and the celebrity Show Lo's scandal.

Moreover, the show's use of puns is specific to a media-based mode of speech in Taiwan, *ganhua* (幹話; lit. trans. words that make me want to curse), which went viral in 2015. The term was coined for the politician Tsai Chi-fang (*Liberty Times Net* 2017) and later describes all kinds of politicians' language that is "not going anywhere", "not helping anyone", "not right anyhow", and "not true anyway" (Hsu 2023). Later, this mode of speech is appropriated by netizens ironically as a device of political trolling, where they make sharp comments against Taiwan's politicians who produce much unironic *ganhua*.[8] In "Ay! Ractopamine Pigs", Tseng creates digressions with puns as his ironic *ganhua* mode of speech, almost exclusively in response to the unironic *ganhua* of politicians. For instance, when a politician comments that banning the ractopamine pork is discrimination against pigs (*yezhu qishi* 野豬歧視), Tseng inserted a homophonic image from a video game, known as "The Pig Knight" (*yezhu qishi* 野豬騎士), to make fun of the politician's comment. Here,

Tseng's use of popular culture is not only incredibly creative but also signals his political alignment with the Taiwan youth mass that has used grassroots online activism against both the DDP and KMT parties in power since the Sunflower Movement.

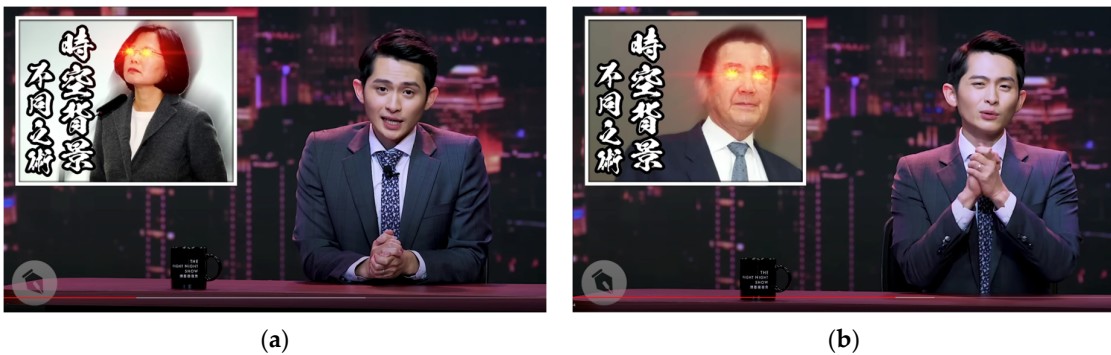

<table>
<tr><td align="center">(**a**)</td><td align="center">(**b**)</td></tr>
</table>

**Figure 4.** Tsai Ing-wen and Ma Ing-jeou are made into matching memes that reference the *Tenet*: (**a**) Tsai is turned into a *Tenet* meme (6:11); (**b**) Ma is turned into the same *Tenet* meme later (11:47).

In contrast to American satire news, what is absent from *The Night Night Show*'s "Ay!" segment is Brian Tseng's satire of the news media itself. Whereas comedians like Stephen Colbert and Jon Stewart habitually make fun of American social media platforms and TV stations for spreading trivial, sensational, and sometimes unreliable news, *The Night Night Show* holds an opposite attitude that is sympathetic and allied with the sensationalist Taiwan media. In the documentary of the Live Comedy Club's members (Luo 2019), Tseng lucidly recognizes the hardship Taiwan's media practitioners face, mainstream and independent alike: "Well the political shows… All business sponsors are cowards. Nobody dares to put money in. So, then, [the media platforms] must, MUST go through great troubles" (25:31). For this sympathy, perhaps, Tseng avoids bringing media platforms under attack even in an "Ay!" segment about fake news. Rather, *The Night Night Show* works with the TV news's sensationalist aesthetics, such as its bold subtitles, its flashy transitions, and its broadcasts of fights and bickers among politicians, to heighten its own comic effects. For one, in "Ay! Ractopamine Pigs", the show took its last pun and meme directly from a screenshot of the CTi news report. There, Tsai Ing-wen is dressed as a Qing Dynasty empress, and the subtitle goes, "The world has changed when I woke/ractopamine up!" (*xinglai* 醒來/"萊") (Figure 5).

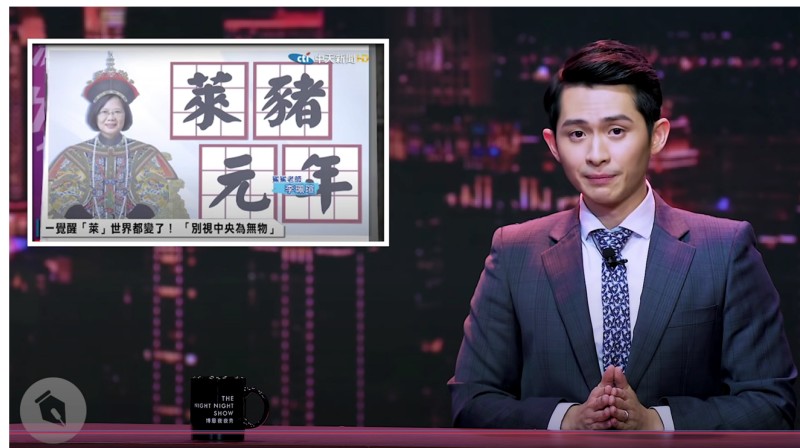

**Figure 5.** Brian Tseng directly uses a screenshot from the CTi news report, which made Tsai Ing-wen into a meme and a pun, to boost the comic effect of his show (17:56).

Overall, *The Night Night Show*'s in-depth presentation of news topics and its explicit engagement with youth culture in Taiwan turns the standards of evaluating satire news'

intervention in truthiness toward a direction different from the US-based studies. In its presentation of truth, *The Night Night Show* for the most part excels in making thoughtful critiques about the systemic flaws in party politics in Taiwan beyond an attack on one or several political personalities. Still, its constructive criticism of political ills is at times lacking. In this episode on the ractopamine pork incident, for instance, the show could have raised more questions about the US's political role in pressuring the government in Taiwan to change its importation policies, instead of generalizing that to the geopolitics of the global market. Additionally, it could have reflected more critically on the news media's role in being complicit with or encouraging the politicians' performative politics. Nevertheless, before being judged as the show's deficiencies in offering constructive critique, these potential weaknesses in *The Night Night Show* should also be put in relation to the show's priority of critical intervention.

As a new format that exists within the neoliberal but necessary sensationalist media environment and responds to a new variety of nationalism that decidedly breaks from the old, *The Night Night Show* intervenes in politics foremostly via reformulating the presentation of truthiness, rather than the presentation of truth. In an interview, *The Night Night Show*'s producer DJ Hauer said that the show expressed what was "truer than news" (*AC-CUPASS* 2018). However, "truer" here means not only the well-reasoned truth that can be evidenced but also the show's ability to draw out the audience's authentic and unfiltered reactions to social ills. The latter interpretation is repeated in Tseng's opening lines of the pilot episode (Tseng 2018): "What does a good social satire talk show mean? It involves an opinionated youth, fresh into the society for only two years, telling you what you should be angry about" (02:45). Again, in this opening line, the empirical wisdom that is associated with age and experience cedes to the burst of anger that is associated with actions and youth.

In both DJ Hauer and Tseng's statements, the satirist's appeal to truthiness is no longer an initiating gesture for the viewers to rationally challenge how social issues have been presented in mainstream news media. Rather, it is an affective mobilization of feelings, passions, and care for news media and social issues alike in the first place, so that people are no longer feeling alienated from or indifferent toward them. For the media theorist Brian Massumi, the affective differentiates from what he calls "emotions"— the sociolinguistic fixing of unique and personal experiences into conventions and consensual points—in that affect is a potentiality (Massumi 2002). Potentiality can be equally intense sensation-wise, but it cannot be fixed into existing social or political categories, where "fixing" is a hegemonic move that always already validates those categories. For example, sadness is a negative "emotion" by social consensus. However, it can also be a pleasant feeling if one is watching a tragic film. Through this lens, *The Night Night Show*'s adaptation of the satire news format can be alternatively understood as presenting a range of potential criticisms but without exhausting all possibilities of critical interactions. Moreover, more important to this affective turn of truthiness is *The Night Night Show*'s innovations in form rather than content. The show is radically political in the densely packed pop culture references, the witty combination of novel and common satirical techniques, and even the inappropriate jokes that offend more than they inform, which aim at maximizing the anti-hegemonic intensity of attraction and shock value. In other words, the critical edge of *The Night Night Show*'s satire is all about recalibrating passions rather than reasons among its audience.

## 5. Conclusions: Brian Tseng's Troubles with "The Line"

On 19 January 2020, after three seasons and a total of 32 episodes, *The Night Night Show* posted its last episode on the STR Network's YouTube channel. While the show was born as a grassroots initiative of standup comedians bringing their live performances online, the show's conclusion ironically came full circle in that it was taken down by what made it—an incident of Brian Tseng's live performance. In August 2019, a user posted on Twitter condemning a joke Brian Tseng made at an unaffiliated open mic event the previous week (*Liberty Times Net* 2019). The poster said a friend at the event recounted that Tseng brought

up Cheng Nan-jung, one of the earliest public advocators for "Taiwan's independence" in the 1980s who committed suicide by self-immolation in support of freedom of speech: "When you burn something in the world of the living, the world of the dead gets the same thing. So ever since Cheng Nan-jung burned himself up, there must have been two Cheng Nan-jung in the underworld!" The butt of the joke was not Cheng Nan-jung but the funeral practice and superstitions, and the audience knew. However, the criticism of Brian Tseng held that using the death of a still-controversial historical figure to make an offhand joke had crossed the line, especially since Cheng is deemed a hero and martyr by many people. In response, the STR Network posted on its Facebook page on August 6, announcing its suspension of Brian Tseng to investigate the controversy (STR Network 薩泰爾娛樂 2019b). Three days later, the Chairman and CEO of the STR Network DJ Hauer posted in defense of Brien Tseng, citing freedom of speech and that the open mic was supposed to be a safe venue for testing jokes. Almost simultaneously, however, Brian Tseng posted on his Facebook that he felt sorry for the negative impact his performance had caused in society, and most importantly, for tearing up the "safety net" of open mic (STR Network 薩泰爾娛樂 2019a). For these reasons, he had reached an agreement with the STR Network that he would step down as the host of *The Night Night Show* and would hand the fourth season to someone else (Stand up 2019). As it resolved, Brian Tseng ended *The Night Night Show* permanently after hosting its last episode in the third season.[9]

The Cheng Nan-jung controversy was neither the first nor last time that Brian Tseng was accused of crossing the line with his off-handed jokes. At the outset, angry netizens cited Tseng for his previous "hellish gags" (*diyu geng* 地獄梗; i.e., "jokes that use other people's disability, tragedy, illness, racial discrimination, war, occupation, death, etc., as a point of humor" (Li 2023) made in *The Night Night Show* and earlier standup performances. In his recap video (2019), the YouTuber Shasha Simpleinfo (*Zhiqiqiqi* 2019) correctly observed that the Cheng Nan-jung controversy was only an excuse that released some viewers' long-unspoken discomfort with Tseng's humor, the risks of which often outweighed rewards. One year after the conclusion of the show, when the STR network posted the unpublished clip of "Ay! Ractopamine Pigs" as a free-standing video, another round of criticism resurged but from a different group of the audience: Tseng's unwanted viewers from mainland China who bypassed the Great Firewall. Knowing the ractopamine pork policies of the ROC government had little to do with the PRC, the unwanted mainland viewers did not come after Tseng's Xi meme but found his joke about exporting the ractopamine pork floss to feed mainland people to have crossed the line. Their anger was directed at Tseng for coming up with this joke and trying to gloss over it by saying "Of course we can't do that!" afterward, but more so at Tseng's audience in Taiwan for cheering and clapping after that idea.[10] The pork floss controversy eventually led to the Taiwan Affairs Office of the State Council of the PRC making a statement that acknowledged "some netizen from the [Taiwan] Island proposed some vicious idea", in addition to announcing that "relevant mainland authorities have stepped up their inspections" on meat products from Taiwan (Zha et al. 2021).

In both cases, Brian Tseng came under fire for "crossing the line", except the "line" here is constantly shifting. The Cheng Nan-jung joke did not become the pressure point for Taiwan's local viewers to break their silence because it was more "hellish" than his other "gags". Instead, it made fun of the political identity with which most of Tseng's audience identified and which has been consistently promoted by Tseng: a civic nationalism that Cheng Nan-jung represents. On the mainland viewers' part, however, most people supported Tseng because they merely read this controversy as a comic failure, where a comedian accidentally stepped on his own toes by making fun of a freedom-of-speech martyr that he should have admired.[11] When the ractopamine pork controversy took place, likewise, the anger of the mainland viewers never crossed the Taiwan Strait, despite its impact in mainland China. The comment section under "Ay! Ractopamine Pigs" is overwhelmingly positive, filled with messages about how much the audience missed the show. No one seems to have noticed his joke about exporting pork floss to mainland China at all.

Together, the controversies show that there does not exist such a hard "line" of political correctness or morality that Brian Tseng cannot cross, because where the line lies depends on the audience's personal, moral, and political alignment. That the controversies were partly responsible for the show's termination nonetheless exposed the fragility of Brian Tseng in *The Night Night Show*: to what extent could the show's radical shock value sustain its existence without having to reasonably address the implicit personal, moral, and political alignments (i.e., his non-satirical faith in the US model as the alternative language for a new Taiwan, his solidarity with Taiwan's news media amid the "media disorder", and his commitment to the civic nationalism that is both engaged in non-partisan activism and anti-Sinoist sentiments)?

On this note, I conclude by highlighting some of *The Night Night Show*'s strengths while leaving those critical issues open for future discussions. First, as an example of the American satire news format outside of the US, *The Night Night* Show's adaptation of the format demonstrates how satire can dismantle linguistic and national boundaries as a transnational bonding force. Yet, it does so not without engaging and highlighting the creative and political agency of Taiwan's comedians as the mediators, who both update the format and alter it to speak to the specific historical, social, and political demands of Taiwan. Second, *The Night Night Show* critically embedded itself in the aesthetic of sensationalism. By mobilizing the mainstream media aesthetics and tropes, it disrupts the binary assumption that in the political discourse, sober truths and inflammatory passions are antagonistic to each other. Lastly, the show rejects the possible belief that the American satire news format universally problematizes the public rhetoric of truth as its primary interest. From a Taiwan-centered perspective, *The Night Night Show* demonstrates how accessing truth not only needs rational and critical thinking but also tuned-in affective responses. Consequently, *The Night Night Show* with Brian Tseng has left a rich legacy for future Sinophone satire media to work with although the show itself has come to an end.

**Funding:** This research received no external funding.

**Institutional Review Board Statement:** Not applicable.

**Informed Consent Statement:** Not applicable.

**Data Availability Statement:** Not applicable.

**Conflicts of Interest:** The author declares no conflict of interest.

## Notes

[1] The martial law period here refers to the 38 years from 1949 to 1987, beginning with the year that the Republic of China (ROC)'s government retreated to Taiwan during the Chinese Civil War (1945–1949), and ending with the end of the global Cold War as well as the rise of the Democratic Progress Party (DPP) that destabilized the political dominance of the Chinese Nationalist Party (KMT) in Taiwan.

[2] Anti-Sinoism is also linked (and sometimes synonymous) with as a pro- "Taiwan's Independence" sentiment that rejects the "One China, different interpretations" system from the 92 Consensus between the PRC government, represented by the Chinese Communist Party (CCP), and the ROC government, then represented by the KMT.

[3] Ho argues that one key condition that allowed activists' transition to party politics is that the movement was driven by "citizens' yearning for genuine democracy, not by disillusionment or alienation". However, Au observes this civic nationalism also risks being absorbed by identity politics that is based on antagonism toward China, instead of positive cultivation of an indigenous democratic culture.

[4] Crosstalk combines storytelling, singing, and speaking in mostly northern Chinese topolects and can be performed in duals, in groups, or in solo formats. In 1949, when the ROC government retreated to Taiwan, most crosstalk masters stayed in mainland China with a few exceptions. After the art form was brought to Taiwan, its impact and practitioners dwindled throughout the martial law period. At the lifting of martial law, however, performance groups were formed in an experimental spirit to revive and update crosstalk in Taiwan.

[5] Notably, this aphasia sentiment is not unique to the comedians in Taiwan within the Sinophone world. As early as April 2016, a late-night talk show program *edu Liang Huan xiu* (恶毒梁欢秀; lit. trans. The Malicious Liang Huan Show), streamed on a major independent online streaming platform Sohu TV in mainland China, already introduced the American satire news format from Comedy Central to the Sinophone audience. The show only lasted for two seasons despite its popularity. Still, its appearance

shows how the Sinophone world's acceptance of the American format is not entirely based on ideological alignment. Rather, it should be interpreted as an alternative language to articulate cultural, or even civic identity, especially when the established ones were put under question.

[6] Translations are mine. "The Awkward Copycat Show" corresponds to "尷尬抄襲秀" (*ganga chaoxi xiu*; lit. trans. Awkward Plagiarism Show) in the source text.

[7] In its final season in 2019, the standup comedian Acid (酸酸), also from the Live Comedy Club Taipei, had a returning in-character skit segment within *The Night Night Show*. The comedian Hello also hosted a returning segment "Hello Night Show" that made a cheaply produced imitation of Brian Tseng's "Random News Report" segment within *The Night Night Show*.

[8] For example, the Taiwan political satire YouTuber *Juruo* (蒟蒻) named his channel *Juruo jiang ganhua* (蒟蒻講幹話; lit. trans. Konjac speaks *ganhua*) with the term "*ganhua*".

[9] On 2 October 2023, the STR Network channel announced the return of *The Night Night Show*, but with Hello as its new host. At this article's time of writing, *The Night Night Show with Hello* is projected to film its first episode on 28 October 2023. The STR Network has not announced to what degree the returned *The Night Night Show with Hello* will continue its previous format.

[10] Whereas YouTube is banned from mainland China, many independent channels on mainland China's main video-sharing platform Bilibili have reposted the clip of Brian Tseng commenting on exporting ractopamine pork floss to mainland China. For example, see the comment section of the clip "台湾网红称：进口莱猪，做成肉松，卖给大陆" [An Influencer from Taiwan Said: Import Ractopamine Port, Make It Into Pork Floss, And Sell It to Mainland China] (Bilibili, https://www.bilibili.com/video/BV1Kz4y1D7Z7/?spm_id_from=333.788.recommend_more_video.-1&vd_source=f1a8c05579aa5c49495d54e78f53331d; accessed 29 September 2023).

[11] The original thread on Zhihu.com have been deleted but is preserved in this video: Shasha Simpleinfo 志祺七七, "中國知乎是如何評價Youtube博恩夜夜秀的？【七七日常】" [How does the Chinese *Zhihu* Comment on The Night Night Show with Brian Tseng on Youtube? (Shasha Daily)] *Qiqi richang* (七七日常), 7 March 2020. Accessed 29 September 2023 https://www.youtube.com/watch?v=O37ELRZoca0&ab_channel=%E4%B8%83%E4%B8%83%E6%97%A5%E5%B8%B8.

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
