# Peer review of "Satirizing News Media, Changing Taiwan’s Feelings: The Night Night Show with Brian Tseng’s Adaptation of the American Satire News Format"

_journalmedia, doi:10.3390/journalmedia4040070_

Round 1
Reviewer 1 Report
Comments and Suggestions for Authors
This article is a great contribution to the proposed special issue. Dialog between Western and Asian comedy is necessary, and the present article provides valuable expertise and a compelling argument. To look at how local adaptations of the satirical news format contribute to the genre given their specific political, social, economic, and cultural contexts is a necessary perspective that counters, in addition, US-centered, pseudo-imperialist arguments about domination of comedic formats.
However, I suggest it be reworked slightly, precisely to strengthen some of its central claims: for one, truthiness seems to want to figure prominently but is entirely underelaborated; there is no proper context for the US's satirical news shows to validate the article's sometimes-comparative readings. Moreover, there is a fascinating point about the role of affect and emotions in political satire that is, however, not a prominent enough part of the article's argumentative or theoretical foundations. References to emotions also suffer from incoherent terminology (sensations, emotion, feelings,...).
Therefore, while sections 2 and 3 are comparatively long, section 4--which finally talks about Tseng's show--feels not long enough. Sections 2 and 3 are rightfully dedicated to offer an enormous amount of context, both in terms of Taiwan's political and comedic histories and developments; it feels at times as though the article wants to provide a comparative history of Taiwan, China, and the US, which it cannot do here, of course. It might be necessary to cut some of these sections in order to really focus on the central argument, which is centered around Tseng's show and its strategies of adapatation AND surpassing the originals. In order to strengthen this red herring, I suggest to move around some really good paragraphs / topic sentences and to not leave it to the end of sections to state the central claims. Suggestions in the document.

Comments on the Quality of English LanguageThe prose in this article flows nicely and is engaging. Sometimes, there is a tendency to make overly long sentences, or to produce verbiage; a careful edit is necessary to spot minor mistakes in punctuation and citation.
Reviewer 2 Report
Comments and Suggestions for Authors
This is an excellent paper that vigorously contextualizes Taiwan news media and offers, to date, the most detailed examination of local adaptions of the American satire news format. I hope the editors will accept and publish it in the present form.
Author Response
Thank you very much for taking the time to review this manuscript and for your recommendation for its publication. I truly appreciate your words of kindness.